# Understanding the Genome-Wide Transcription Response To Various cAMP Levels in Bacteria Using Phenomenological Models

Shweta Chakraborty,[a] Parul Singh,[b] Aswin Sai Narain Seshasayee[a]

[a]National Centre for Biological Sciences, Tata Institute of Fundamental Research, Bangalore, Karnataka, India
[b]Institut Pasteur, Paris, France

**ABSTRACT** Attempts to understand gene regulation by global transcription factors have largely been limited to expression studies under binary conditions of presence and absence of the transcription factor. Studies addressing genome-wide transcriptional responses to changing transcription factor concentration at high resolution are lacking. Here, we create a data set containing the entire *Escherichia coli* transcriptome in Luria-Bertani (LB) broth as it responds to 10 different cAMP concentrations spanning the biological range. We use the Hill's model to accurately summarize individual gene responses into three intuitively understandable parameters, $E_{max}$, $n$, and $k$, reflecting the sensitivity, nonlinearity, and midpoint of the dynamic range. Our data show that most cAMP-regulated genes have an $n$ of >2, with their $k$ values centered around the wild-type concentration of cAMP. Additionally, cAMP receptor protein (CRP) affinity to a promoter is correlated with $E_{max}$ but not $k$, hinting that a high-affinity CRP promoter need not ensure transcriptional activation at lower cAMP concentrations and instead affects the magnitude of the response. Finally, genes belonging to different functional classes are tuned to have different $k$, $n$, and $E_{max}$ values. We demonstrate that phenomenological models are a better alternative for studying gene expression trends than classical clustering methods, with the phenomenological constants providing greater insights into how genes are tuned in a regulatory network.

**IMPORTANCE** Different genes may follow different trends in response to various transcription factor concentrations. In this study, we ask two questions: (i) what are the trends that different genes follow in response to changing transcription factor concentrations and (ii) what methods can be used to extract information from the gene trends so obtained. We demonstrate a method to analyze transcription factor concentration-dependent genome-wide expression data using phenomenological models. Conventional clustering methods and principal-component analysis (PCA) can be used to summarize trends in data but have limited interpretability. The use of phenomenological models greatly enhances the interpretability and thus utility of conventional clustering. Transformation of dose-response data into phenomenological constants opens up avenues to ask and answer many different kinds of question. We show that the phenomenological constants obtained from the model fits can be used to generate insights about network topology and allows integration of other experimental data such as chromatin immunoprecipitation sequencing (ChIP-seq) to understand the system in greater detail.

**KEYWORDS** CRP, *E. coli*, Hill's model, LB, phenomenological constants, cAMP

Transcription of a gene is the result of a coordinated effort between transcription factors, various nucleoid-associated proteins (NAPs), and the RNA polymerase (RNAP) at the promoter of a gene. In bacteria, this process is tightly regulated, allowing rapid adaptation and

Address correspondence to Aswin Sai Narain Seshasayee, aswin@ncbs.res.in, or Shweta Chakraborty, shwetachakraborty09@gmail.com.

The authors declare no conflict of interest.

10.1128/msystems.00900-22 1

survival in changing environments (1–3). Most studies that focus on understanding the role of a transcription factor in the cell use knockout strains to study the changes in gene expression in the presence and absence of the transcription factors. The advent of high-throughput RNA-sequencing (RNA-seq) techniques in knockout strains has accelerated our understanding of the roles of various transcription factors in maintaining the optimum physiology of the cell (4–10). Such experiments give us both qualitative (which genes are differentially expressed) and quantitative (magnitude of the change) information.

Concentrations of many transcription factors may exhibit a switch-like behavior. Their effects may be well modeled by binary states of the transcription factor (11, 12). However, there are many regulators whose concentration in the cell varies continuously (13–16). While studies under binary states of transcription factor concentrations capture snapshots of the gene expression changes, they fail to give any information about the various response of different genes to changing concentrations of the transcription factor. However, across the genome, different sets of genes in the cell may be tuned to respond differently to changes in transcription factor levels (17–19). In an attempt to better understand the regulatory mechanisms of various transcription factors, previous studies have focused on gene expression changes in response to modulating transcription factor concentrations (15, 20–23). However, these studies have been limited to a few genes or low resolutions of the regulator concentration. In the recent past, Yang et al., using microarrays, analyzed trends in global gene expression patterns across seven different cAMP concentrations. Overall, the literature for such studies on a global scale has remained sparse (15, 23, 24).

One example of a transcription regulator whose concentration in the cell varies continuously is the cAMP signaling system in *Escherichia coli*. cAMP is well known for its role in carbon catabolite repression (CCR) and hierarchical utilization of carbon sources (19, 25–28). cAMP exerts its effect on gene expression by binding and activating the transcription factor cAMP receptor protein (CRP) (5, 29, 30). Over years, cAMP along with its effector CRP have been shown to have pleiotropic roles in carbon and nitrogen metabolism, motility, biofilm formation, and survival against various stress conditions (8, 31–36). Recent studies have implicated cAMP as a key player in optimizing proteome allocation and maintaining flux balance across diverse nutritional conditions (37–39). Intracellular cAMP concentrations are regulated by the nutritional state of the cell. Carbon sources that support high growth rates inhibit cAMP production. As the cell experiences worse nutritional conditions, intracellular cAMP concentrations increase proportionately. Increasing cAMP levels redirect greater cellular resources toward the expression of transport and metabolic genes (28, 40). *E. coli* cells experience cAMP changes across carbon sources (21, 41), and cAMP concentrations in the cell vary continuously in response to cell density and growth phases (42–44).

In *E. coli*, the cAMP-CRP complex is responsible for regulating more than 500 genes, across different conditions (34, 40, 45). Because cells experience a continuous change in cAMP concentrations, we wanted to quantitatively characterize the different trends each gene may follow in response to activation by cAMP. Previously, Liu et al. qualitatively described the response of cAMP-regulated genes to increasing cAMP concentrations achieved by altering carbon sources (21). Other works have shown that it is possible to control activities of cAMP-regulated genes by modulating extracellular cAMP, allowing easy experimental control over this system (22, 24, 33, 46, 47).

The enzyme adenylate cyclase, encoded by the *cyaA* gene, is the sole source of cAMP production in *E. coli* (48). In this study, we expose cAMP-deficient Δ*cyaA E. coli* mutants to 10 different concentrations of extracellular cAMP and measure the global gene expression changes in response to various cAMP concentrations using RNA-seq. We use phenomenological models to quantitatively describe the different trends that individual genes follow in response to increasing levels of cAMP. We show that most cAMP-responsive genes follow a sigmoid curve, which can be satisfactorily modeled using Hill's equation. The presence of a sigmoid-type response curve in gene expression circuits is not uncommon. As a matter of fact, the prevalence of feedback and feedforward loops in regulatory networks make the sigmoid response motif extremely common. Further, the use of the Hill class of equations to

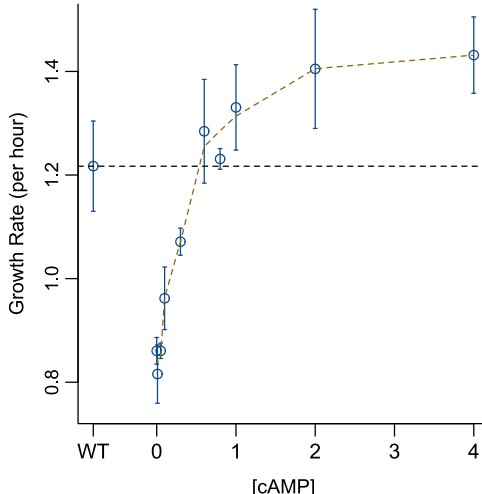

**FIG 1** Changes in growth rate with increasing cAMP dosage. Δ*cyaA E. coli* cells exhibit a growth defect compared to the wild-type strain when grown in LB media. The 0 mM cAMP concentration represents the Δ*cyaA* strain in LB with no added extracellular cAMP and WT represents the wild-type strain. Growth rates of Δ*cyaA* cells increase monotonically in response to increasing extracellular cAMP concentrations. Each point represents the mean of three biological replicates with error bars representing the standard deviation around the mean. Growth curves for the wild-type and Δ*cyaA* mutant growing in LB are shown in Fig. S1A in the supplemental material. The horizontal dotted line represents the wild-type growth rate.

model dose-response curves has been found to be quite useful in understanding interaction kinetics in biochemical and pharmacological studies. In the recent past, it has been found to be useful in the modeling of transcription regulatory networks as well (49–54).

In the next part of the paper, we try and explore the use of the phenomenological constants obtained from Hill's model (HM) to gain insights into the workings of the cAMP regulatory network. In this study, we use the four-parameter Hill's model defined by $b0$, $n$, $k$, and $E_{max}$. The biological implications of these constants have been hashed out for single transcription factor-gene interactions (50, 52, 55–57). We attempt to use these parameters to study the properties of the cAMP regulatory network.

## RESULTS

**Effects of increasing cAMP concentrations on growth rates of the Δ*cyaA E. coli* mutant.** Adenylate cyclase is encoded by the *cyaA* gene in *E. coli* and converts ATP to cAMP. Δ*cyaA E. coli* mutants cannot make intracellular cAMP (29, 58). *E. coli* cells that lack the *cyaA* gene are capable of growing in rich permissive media, like Luria-Bertani (LB) broth, albeit with a growth defect (Fig. S1A and B in the supplemental material). However, these mutants exhibit a complete loss of growth when growing in sugars such as lactose or a sorbitol-ribose mixture that require the cAMP-CRP signaling system for their utilization (Fig. S1C and D).

We wanted to see if the addition of extracellular cAMP can rescue the growth defect induced by *cyaA* deletion. To this end, we studied the growth dynamics of Δ*cyaA* cells across 10 different extracellular cAMP concentrations ranging from 0 mM to 4 mM cAMP in three different media: LB, M9 with lactose, and M9 with a mixture of sorbitol and ribose. To determine if extracellular cAMP enters the cell, we measured the levels of intracellular cAMP at each administered dose. We found a linear relationship between the measured intracellular cAMP and extracellular cAMP provided (Fig. S1G). For the rest of the study, all instances of cAMP refer to the extracellular concentrations provided, unless mentioned otherwise.

As expected, Δ*cyaA* (0 mM cAMP) cells showed a lower growth rate in LB than the wild-type (WT) cells (Fig. 1). In M9 medium with sugar, Δ*cyaA* cells failed to grow in the absence of cAMP (Fig. S1C and D). Irrespective of the medium, exposure to various concentrations of cAMP induced an increase in growth rates in a monotonic fashion,

with growth rates being restored to wild-type levels at high cAMP concentrations between 0.8 and 4 mM cAMP (Fig. 1 and Fig. S1C and D). In LB, growth rates reached wild-type levels between 0.6 and 0.8 mM cAMP. We measured the concentration of cAMP in wild-type *E. coli* cells growing in LB to be 5.4 ($\pm$1.06) pmol/optical density (OD)/mL (Fig. S1B). The concentration is equivalent to 0.72 ($\pm$0.12) mM extracellular cAMP, which is close to the administered cAMP concentration at which the $\Delta cyaA$ cells reached wild-type levels. We observed that no change in growth rates was induced until 0.3 mM cAMP. This could be because the amount of cAMP entering the cells at low concentrations was insufficient to affect a transcriptional response.

One difference in growth dynamics for cells growing in M9 minimal medium with sugars compared to LB was in their lag times. In batch culture, lag time refers to the time taken by a population to exit the lag phase and transit to the log phase of growth. In LB, the $\Delta cyaA$ cells showed no change in lag time (1 h) in response to the cAMP concentration gradient. However, cells cultured in lactose and sorbitol-ribose media experienced decreasing lengths of lag time with increasing cAMP concentrations (Fig. S1E and F).

**Global transcriptional changes in the $\Delta cyaA$ mutant in response to ordered exposure of cAMP.** The cAMP-CRP signaling system in *E. coli* is a global regulator of transcription. Because the addition of extracellular cAMP was able to restore the growth rates of $\Delta cyaA$ mutants to wild-type levels, we wanted to see if it had the same effect on the transcriptome as well. We grew $\Delta cyaA$ cells in LB with 10 increasing doses of extracellular cAMP followed by whole-transcriptome RNA sequencing at each cAMP concentration.

cAMP and CRP have been reported to regulate more than 500 genes in *E. coli* across various conditions (5, 6, 40). In LB with no added cAMP, we found 488 genes to be differentially expressed between wild-type and the $\Delta cyaA$ mutant (log$_2$ (Fold Change) > $\pm$1 and $P$ < 0.01); 305 genes were upregulated in the wild-type strain compared to the $\Delta cyaA$ mutant, while 183 genes were downregulated. For any given gene, we calculated its response to a cAMP concentration as the fold change in expression experienced by the gene at that cAMP concentration compared to the 0 mM cAMP state ($\Delta cyaA$). We observed that genes under cAMP control responded monotonically to increasing concentrations of extracellular cAMP, with positively regulated genes expressing and negatively regulated genes repressing monotonically with increasing levels of cAMP (Fig. 2A and B). Because cAMP-CRP is primarily a transcriptional activator, in this study, we concentrated only on genes that are under positive regulation by cAMP.

Consistent with our observations in the growth rate studies in LB, we observed changes in gene expression only after 0.3 mM cAMP, followed by a monotonic increase in expression (Fig. 2C). A large number of genes expressed between 0.6 mM and 0.8 mM cAMP, beyond which changes in gene expression started saturating (Fig. S2C). We observed that 0.8 mM cAMP was sufficient to induce wild-type levels of growth rates in $\Delta cyaA$ cells (Fig. 1) but not sufficient to restore the expression of all genes to that of wild-type levels (Pearson correlation, $r$ = 0.79, slope = 0.73, $P$ < 0.01; Fig. 2C). Despite some genes failing to reach wild-type levels, gene expression levels at 0.8 mM cAMP were sufficient to gain wild-type growth rates, indicating that all differentially expressed genes did not contribute equally toward the maintenance of cellular growth rates.

A strong correlation of 0.93 with a slope of 1 ($P$ < 0.01) between gene expression at 2 mM cAMP and 4 mM cAMP revealed that expression saturated by 2 mM cAMP (Fig. 2D). By 2 mM cAMP, the transcriptome was also well correlated with that of the wild-type (Pearson correlation, $r$ = 0.87 and slope = 0.84, $P$ < 0.01). However, even at high concentrations of extracellular cAMP, not all cAMP-regulated genes reached wild-type levels of expression, with the expression of many genes remaining lower than that of the wild-type (Fig. 2D, yellow points). Despite administering doses of cAMP 4 times that of the wild-type concentrations, only 10 genes were differentially expressed between the 4 mM cAMP and wild-type groups (Fig. S2B).

The cAMP-CRP complex binds to promoters and recruits RNA polymerase (RNAP), facilitating the transcription of a gene. We wanted to study the effects of CRP or RNAP binding at the promoter on gene expression in the wild-type cell. We used signals

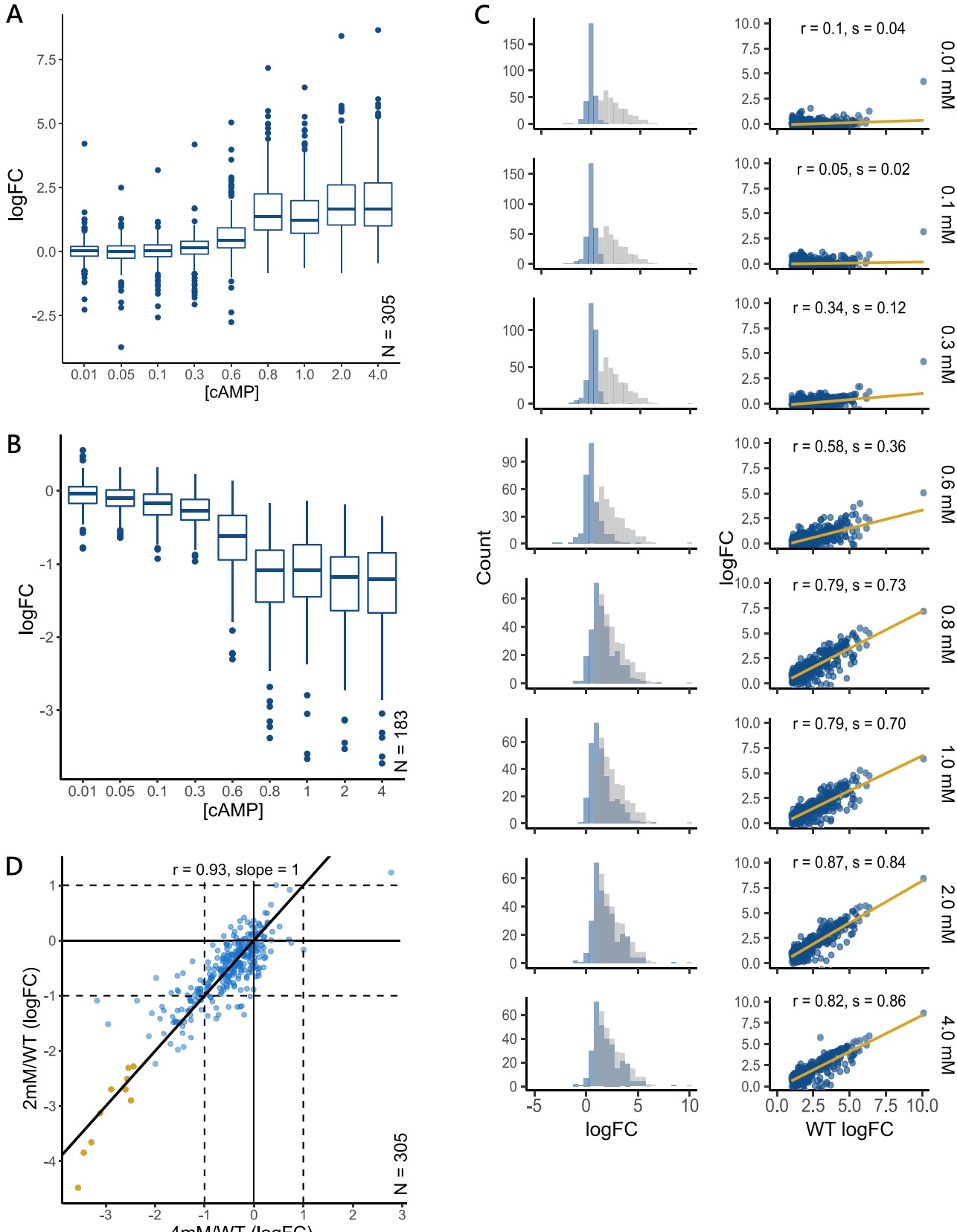

**FIG 2** Expression changes of the cAMP regulon in response to increasing cAMP concentrations. (A and B) Distribution of gene expression values at different doses of extracellular cAMP for genes under positive (A) and negative (B) regulation of cAMP. In LB, 305 genes are positively regulated and

obtained from chromatin immunoprecipitation (ChIP) studies to calculate the occupancy of CRP or RNA polymerase at a promoter. For CRP, we used ChIP-seq to measure the CRP occupancy across the entire chromosome in exponentially growing wild-type cells. For RNAP, publicly available ChIP data (59) were used to calculate the RNAP occupancy at the promoter of each gene or operon (Methods 5). The median of the distribution for both CRP and RNAP occupancy at promoters of genes that are positively regulated by cAMP was significantly higher than that of nondifferentially expressed (non-DE) genes (Wilcoxon rank-sum test, $P < 0.001$; Fig. S3A and B). Further, we found a significant correlation of 0.46 ($P = 4.14 \times 10^{-12}$) between the fold change in expression (wild-type with respect to the $\Delta cyaA$ mutant) of genes positively regulated by cAMP and their CRP occupancy. However, no such correlation was observed for RNAP occupancy and expression (Fig. S3C and D).

Given that cAMP activated close to 300 genes in LB, studying variations in individual gene expression kinetics in response to cAMP can give deeper insights into the nature of the cAMP-CRP regulatory network. To this end, we attempted to characterize the various trends that genes may follow in response to the ordered exposure of cAMP.

**Use of phenomenological models to describe cAMP-regulated genes.** In the previous section, we quantified changes in gene expression induced by cAMP exposure. To generate dose-response curves for each gene, we calculated the effect of cAMP concentration on a gene as the change in expression (fold change) of the gene at that cAMP concentration relative to its expression in the $\Delta cyaA$ mutant treated with 0 mM cAMP. To classify and quantitatively summarize these dose-response curves, we made use of phenomenological models. Phenomenological models are a useful tool to quantitatively describe trends independent of the underlying mechanisms that produced them. Further, the phenomenological constants obtained from these models can be used for characterization of the observed behavior.

Dose-response curves often follow a sigmoid behavior. Such nonlinear behavior of gene expression in response to activation by transcription factors is well known in biological systems. The presence of feedforward loops in regulatory networks and cooperative behavior of transcription factors in enhancing their own binding at the promoter lead to sigmoid behavior (56, 60–62). Hill-type models best describe such sigmoid trends. We used the four-parameter Hill's model, defined by $b0$, $k$, $n$, and $E_{max}$, to characterize genes following sigmoid behavior in our data (50). It is possible that some genes do not reach their saturation levels under the regimen of the experiment. These would appear to have a nonsaturating trend. We used the linear model to describe such a response. We compared these models against a null model defined by no change in response to the given signal (Methods 6).

Overall, we used three models to bin the dose-response curves: the null model (NR) to identify genes that show no change in expression in response to increasing cAMP concentrations, the Hills model (HM) to fit genes that show sigmoid and first-order Michaelis-Menten behavior, and the linear model (LM) for genes that exhibit nonsaturating behavior within the cAMP range used in this study. Gene trends satisfied the Hill's model (HM) only if the residual sum error ($\sigma^2$) of the fit by HM was less than that of the competing models, its $R^2_{NLfit}$ was greater than 0.8 ($P < 0.05$), the slope$_{NLfit}$ was equal to $1 \pm 0.2$ and relative standard error (RSE) for estimated parameters $k$ and $E_{max}$ was less than 20% (Methods 6). Because the number of data points was less around the transition points, the predictions for $n$ showed a larger variation in RSE. Hence, we did not use the RSE of $n$ as a cutoff factor. Similarly, we considered genes to

**FIG 2 Legend (Continued)**

183 genes are negatively regulated by cAMP in the wild-type strain compared to the $\Delta cyaA$ mutant. For a gene, change in expression was calculated as the $\log_2$ (Fold Change) at a cAMP concentration compared to the 0 mM cAMP $\Delta cyaA$ E. coli strain. (C) Left, frequency distribution of gene expression for positively regulated genes at each cAMP concentration (blue) compared to the wild-type distribution (gray). Right, correlation between gene expression at each cAMP concentration (y-axis) with wild-type gene expression (x-axis). The variables $r$ and $s$ represent the Pearson correlation coefficient and slope, respectively. (D) Scatterplot for correlation between gene expression at 2 mM and 4 mM cAMP concentrations. Each point represents a gene. A strong Pearson correlation of 0.93 ($P < 0.01$) shows that gene expression has saturated by 2 mM cAMP. Some genes are not expressed despite the addition of cAMP as high as 4 mM (yellow).

fit the linear model (LM) if $\sigma^2$ for the fit was less than other models, its $R^2_{\text{NLfit}}$ was greater than 0.7, and the $P$ value was less than 0.05. Genes followed the null model if the $\sigma^2$ for the fit was less than the other models and $R^2_{\text{NLfit}}$ was greater than 0.7 ($P < 0.05$) and slope$_{\text{NLfit}}$ was equal to $1 \pm 0.2$.

Using the model fit method described above, we found that 74% (225/305) of cAMP-regulated genes followed the HM, while 9% (28/305) of genes followed the LM. Despite being differentially expressed in the wild-type strain compared to the Δ*cyaA* mutant, 6% (18/305) of genes fit the null model (NR) and did not show any change in expression in response to extracellular cAMP; 11% (34/305) of genes did not fit any of the above models. These genes showed a nonmonotonic (NM) response to increasing cAMP concentrations (Fig. 3A; Table S1).

Unsupervised clustering is one of the common methods used to study trends in a data set (63–65). We compared the model fit method described above to the results obtained using clustering algorithms, such as *k*-means and hierarchical clustering. Both these methods revealed the continuous nature of the data and the lack of clear clusters in it. This was reflected in the inability of methods meant to find optimum clusters for *k*-means to reach a consensus (Fig. S4A) and the presence of large cross-correlation across clusters in the distance matrix (Fig. S4E). Based on the Silhouette method, we divided the genes into three clusters using *k*-means. The *k*-means algorithm yielded two major clusters containing 31.1% (95/305) and 64.2% (196/305) of the genes and a small cluster with 4% (14/305) of genes. The two major clusters showed a median trend resembling sigmoid curves (Fig. S4B and C).

Because the model fit method showed that cAMP-regulated genes fall into four clusters, we forced the *k*-means algorithm to divide the data into four clusters. This yielded patterns similar to those from the model fit (Fig. 3B). For easier visualization, we overlaid the genes into groups partitioned by *k*-means and model fit on a two-dimensional (2D) principal-component analysis (PCA) plot (Fig. 3C). We observed that gene trends are in a continuum with partitioning by both methods happening primarily across the first principal component (PC1), where gene trends vary from NM to HM to LM. Each *k*-mean cluster primarily corresponded to at least one trend from the model fit (Fig. 3D and E). Clusters generated by the *k*-means algorithm largely agreed with partitions made by the average linkage hierarchical clustering method used to analyze the gene trends (Fig. S5).

In this section we showed that genes under the positive regulation of cAMP broadly follow four trends: sigmoid, linear, nonmonotonic, and nonresponsive. The overall gene expression response to cAMP populates a continuum compared to distinct clusters, with the majority of genes following a sigmoid dose-response curve that could be described quantitatively by phenomenological models such as the Hill's model. We note that conventional clustering methods can in fact meaningfully partition the data into intuitively understandable shapes. However, finding the optimum clusters in continuous data poses a difficult decision. Model fitting circumvents this problem by adding another layer of clarity, making it possible to quantitate the variations observed across these continuous trends. Not only this, parameters obtained from the Hill's model can be used to further characterize the behavior of these dose-response curves.

**Characterization of phenomenological constants obtained from the Hill's model.** For a gene whose transcription is activated by cAMP, the Hill's model defines a sigmoid dose-response curve using four parameters: (i) $b0$ represents the basal level of gene expression when no cAMP signal is observed, (ii) $k$ quantifies the levels of cAMP required for the gene to reach half of its saturating concentration, (iii) $E_{\text{max}}$ quantifies the magnitude of fold change at the saturating concentration, and (iv) $n$ describes the rate of change of expression in response to changing cAMP levels. Together, $n$ and $k$ determine the dynamic range of a gene in response to cAMP (Fig. 4A).

Biologically, cAMP concentrations at $k$ reflect the midpoint of the dynamic range of a gene. Except two genes, all genes showed a $k$ greater than 0.35 mM, with values of $k$ ranging up to 2.6 mM cAMP (Fig. 4B and S6A). This was consistent with the findings from our global gene expression analysis where we observed no change in gene

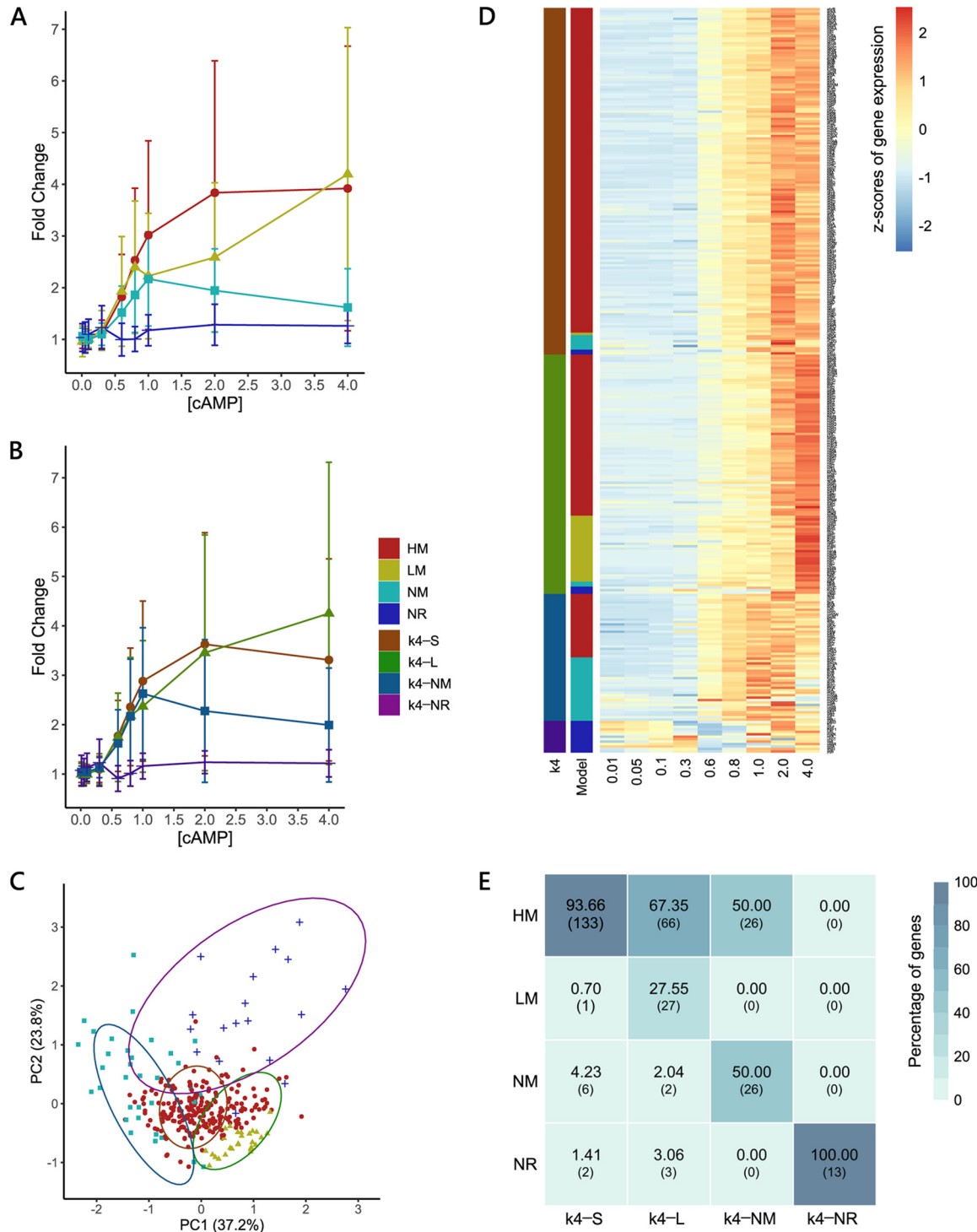

**FIG 3** Comparison of different clustering methods. (A and B) Median trends for clusters from the model fit method (A) and *k*-means with four clusters (B). Each point represents the median expression of the genes belonging to the indicated cluster at that cAMP concentration. Error bars represent the median absolute deviation (MAD) around the median. Broadly, cAMP-regulated genes follow one of four trends: sigmoid (HM/k4-S), linear (LM/k4-L), nonmonotonic (NM/k4-NM), or nonresponsive (NR/k4-NR). (C) Projection of gene expression trends on a PCA plot. Each point represents a gene. Different shapes indicate the cluster a gene belongs to using the model fit method. Ellipses indicate the gene clusters formed using the *k*-means clustering methods. Colors of points and ellipses are consistent with the color key given for (A) and (B). Gene trends are in a continuum, with trends varying smoothly from NM to HM to LM primarily along PC1. (D) Comparison of genes across clusters for the *k*-means and model fit method. Each row represents a gene. For the two clustering methods, colors represent the cluster a gene belongs to. Colors are consistent with the color key given for (A) and (B). The heatmap shows the normalized expression levels for the gene at each cAMP concentration. (E) Correlation plot between genes binned by the model fit and *k*-means with four clusters. Each *k*-means cluster primarily corresponds to at least one trend from the model fit. Numbers in the boxes represent the percentage of k4-X genes that follow the model fit trends with the common number of genes shown in brackets.

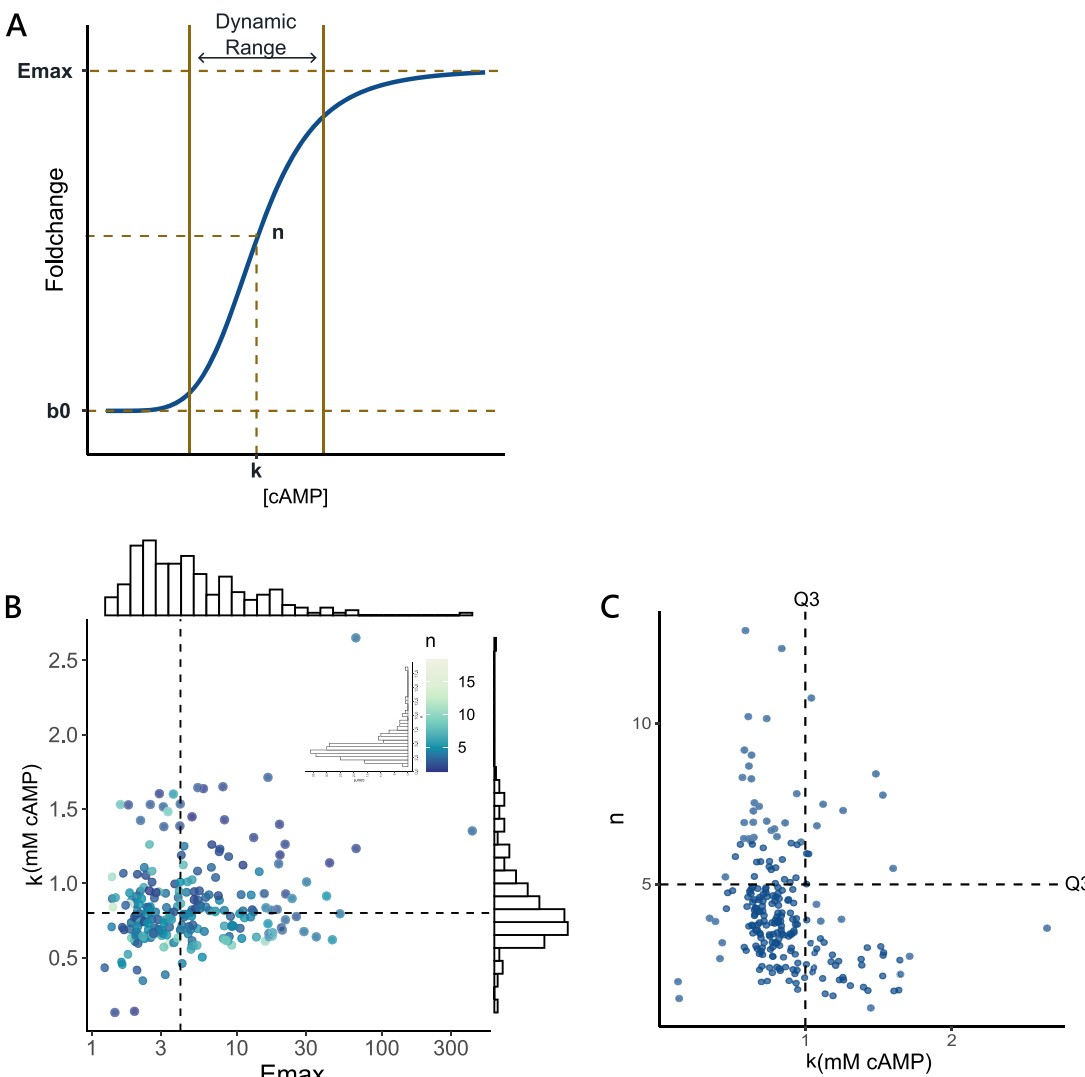

**FIG 4** Distribution of the parameters obtained from the Hill's model. (A) Schematic showing the dose-response curve for a gene following the Hill's model, with the four parameters ($b0$, $k$, $n$, and $E_{max}$) depicted in the figure. $k$ and $n$ together determine the dynamic range of a gene. (B) Distributions of $k$ and $E_{max}$ for genes that follow the Hill's model. The color of each point represents the value of $n$. There is no correlation between $k$ and $E_{max}$ or $n$ and $E_{max}$ (data not shown). Dashed lines show the median of the distributions of $k$ and $E_{max}$. (C) Scatterplot showing the absence of high $k$ and high $n$ genes in the study. Dotted lines represent the value of top third quartile (Q3) for the respective distributions. $N = 225$.

expression before 0.3 mM cAMP (Fig. 1). Genes under cAMP control showed a median $k$ of 0.79 mM cAMP with an interquartile range (IQR) of 0.26 mM. The IQR of $k$ accounted for only 20% of the entire range of $k$, implying that the midpoints of the dynamic range of many cAMP-regulated genes (i.e., 50% of genes) occupy a very small space in the range of $k$. Further, we observed that the median value of $k$ lies close to the wild-type levels of cAMP for *E. coli* growing in LB (0.72 mM cAMP). The observation that the midpoints of the dynamic range of gene expression (median $k$) are centered around the wild-type concentration of cAMP suggests that promoters and intracellular cAMP concentrations in the cAMP-CRP regulatory network may be tuned to have optimized gene expression.

$E_{max}$ defines the maximum expression of a gene when cAMP is not the limiting factor and reflects the sensitivity of a gene to cAMP. We observed that the distribution of $E_{max}$ for cAMP-regulated genes in LB is right skewed, with saturating gene expression varying from 1.2- to 66.5-fold change (except *gatB*, which showed a very high $E_{max}$ of a 420.7-fold change). The variation observed in $E_{max}$ (median absolute deviation from the

median [MADM] = 48.4%) was much greater than that of $k$ (MADM = 16.5%), showing that differences in the $E_{max}$ of genes affected the observed intergene variation in expression more than the differences in $k$ (Fig. 4B and S6C).

$n$ determines the steepness or nonlinearity of the dose-response curve. Nonlinearity in the system is introduced by the cooperative behavior of transcription factors and the presence of positive feedforward and feedback loops. Higher $n$ indicates more switch-like behavior, while low $n$ indicates a more graded response. All genes following the HM exhibited an $n$ greater than 1. $n$ for cAMP-regulated genes showed a median value of 3.9 with an IQR of 2 (Fig. S6B). The high values of $n$ for cAMP-regulated genes reflect the pervasive nature of feedback and feedforward loops in the network, resulting in switch-like behavior for most genes in the population. For the use of this study, we considered values of $n$ greater than 5 to be high, values of $n$ between 3 and 5 to be moderate, and values of $n$ less than 3 to be low.

We observed a lack of genes that had both high $n$ (>5) and high $k$ (>1) in the population (Fig. 4C). Very few genes (such as *torR*, *glpK*, *ybhG*, and *ilvX*) showed both high $n$ and $k$ (Fig. S6D and S6E). This led to an apparent negative relationship between $n$ and $k$. Hence, genes that saturated at high cAMP concentrations (high $k$) were likely to behave in a relatively graded manner (have low $n$) in response to cAMP. Conversely, genes that showed a more switch-like behavior (high $n$) were more likely to also saturate at lower cAMP levels (have low $k$). This pattern could also be the result of the experimental design, as it may be difficult for the algorithm fitting the Hill's model to capture genes with trends having high $k$ and $n$ due to the lack of data points depicting the transition and saturation states at high cAMP concentrations. Although after manually checking the trends of genes rejected by the Hill's model, we noticed that no gene showed such a trend, hinting that the effect may result from biological limits as opposed to experimental limitations.

Overall, our data showed that expression of cAMP-regulated genes in LB differed more in their $E_{max}$ than $k$. Also, instead of having a graded response across the biological range of cAMP, most cAMP-regulated genes exhibited a switch-like behavior, with their dynamic range centered around 0.79 mM cAMP, close to the wild-type concentrations of cAMP in LB. This suggests that cAMP-regulated genes rapidly switch on and reach saturation ($E_{max}$) at concentrations close to 0.79 mM cAMP. This is consistent with the observation that gene expression increases rapidly between 0.6 and 0.8 mM cAMP (Fig. 2C; Fig. S2C).

**Comparison of Hill's parameters across genes under direct and indirect control of cAMP.** The cAMP regulatory network consists of genes that are directly regulated by the cAMP-CRP complex binding at their promoter as well as genes that are under indirect control via intermediate players. We binned the cAMP-regulated genes as direct or indirect targets of the cAMP-CRP complex using two different criteria. In the first classification method, we considered genes to be direct targets of the cAMP-CRP complex if they were also annotated to be directly regulated by CRP in the Ecocyc/RegulonDB database (Direct$_{Ecocyc}$). We binned the rest of the genes as Indirect$_{Ecocyc}$. Of the entire set of cAMP regulated genes 98/305 belonged to Direct$_{Ecocyc}$. For HM genes, 66/225 belonged to Direct$_{Ecocyc}$. The Direct$_{Ecocyc}$ genes showed a greater CRP occupancy at promoters than Indirect$_{Ecocyc}$ genes (Wilcoxon rank-sum test, $P$ = 0.0092; Fig. S7A), confirming that the set of genes are indeed under direct regulation of cAMP. However, we noticed that promoters of genes belonging to Indirect$_{Ecocyc}$ showed greater CRP occupancy scores than nondifferentially expressed genes (Fig. S7A). Because the assumption is that the cAMP-CRP complex only binds to promoters of direct genes, one would expect genes under indirect regulation of CRP to have occupancies similar to those of nondifferentially expressed genes. This indicated that the Ecocyc/RegulonDB-defined set of direct genes may not be complete.

We defined a more inclusive set of Direct$_{ChIP}$ and Indirect$_{ChIP}$ genes based on the data obtained from Ecocyc/RegulonDB database and CRP occupancy scores obtained from the ChIP experiments. The set Direct$_{ChIP}$ was defined by taking the union of Direct$_{Ecocyc}$ and those genes in Indirect$_{Ecocyc}$ that had a CRP occupancy score greater than 2.8. This cutoff was chosen based on the bottom quartile (Q1) of CRP occupancy

scores of the $Direct_{Ecocyc}$ set (Fig. S7B); 147/225 HM genes belonged to the $Direct_{ChIP}$ set. Of all cAMP-activated genes, 211/305 were binned as $Direct_{ChIP}$ (Fig. S7C).

For an activator, $k$ reflects the transcription factor concentration around which the gene behaves most dynamically for cases where $n$ is greater than 1. For a gene that is under direct control of cAMP, $k$ is affected by the affinity of the transcription factor to the promoter. For indirect genes, it reflects the composite effects of all $k$ values in the network. $n$ determines the steepness of the response curve and is affected by the presence of multiple regulators and feedforward and feedback loops present in the circuit. Together, they determine the dynamic range of a gene in response to the transcription factor. Thus, the observed $k$ and $n$ of a gene in the regulatory network may get affected by its level of regulation. Because we did not have any formal expectation of how these parameters may be affected by their levels of regulation, we used a toy model of a simple regulatory circuit to determine how the distributions for these parameters should look across direct and indirect genes.

We considered a linear regulatory chain having four nodes and three edges, with each edge having independent $k$ and following the Hill's function (Fig. 5A). $k$ at each level was chosen randomly from the distribution of observed $k$ from our study. The first edge is akin to the regulation of a direct gene by cAMP. The second edge takes concentrations of the direct gene (g1) as an input function for the first indirect gene (g2, indirect level1). Similarly, the third edge takes the output of the first indirect gene (g2) as an input for the downstream node (g3, indirect level2). We mapped the outcome at each node to the input concentrations of cAMP and calculated the apparent $n$ and $k$ at each level. The starting concentration ($b0$) for each node was set to 0. For this simulation, we did not consider changes in $E_{max}$. $E_{max}$ reflects the maximum fold change a gene can experience when cAMP is unlimited. It is more likely to be affected by promoter-related properties (described in later sections) than the level of regulation. Thus, for simplicity, $E_{max}$ was set to 1 for the toy model. Because all HM genes have an $n$ greater than 1, in this circuit, we fixed the value of $n$ at each edge to 2. We found the mean and variance for distributions of apparent $k$ and $n$ to increase at each regulatory level (Fig. 5B and C). Thus, our null model predicts that the response of genes further down the regulatory circuit show (i) greater nonlinearity ($n$) and (ii) larger variation in the midpoints of their dynamic range ($k$).

Contrary to the null model, our data showed no differences in the distribution of $k$ and $n$ across direct and indirect genes, irrespective of the definition of direct and indirect genes used (Wilcoxon rank-sum test, $P > 0.01$; Fig. 5D and E; Fig. S7D, E and F). Deviation from the null model reflected that either one or both assumptions (linear topology and independent $k$ at each level) of the simple model were violated. We suspect that the presence of feedback loops feeding into higher nodes could cause the variation in $n$ of genes to increase, leading to comparable distributions of $n$ for direct and indirect genes. This finding implies that direct genes are not regulated solely by cAMP and that cyclic edges are pervasive even among direct genes. The presence of cyclic edges feeding into higher nodes could also cause the $k$ values at each step to be dependent on each other, leading to violation of the null model. Another reason could be that the cAMP regulatory network mostly consists of short-length subnetworks limiting the variation in $k$ for indirect genes.

**Contribution of CRP and RNA polymerase binding on gene expression.** A successful transcription event is the product of concerted interactions of various molecular players, such as regulators and helper nucleoid-associated proteins (NAP) with the RNA polymerase (RNAP) at the promoter. Models like the Hill's model can help abstract the strength of these interactions into physiologically relevant phenomenological constants. For cAMP-regulated genes, two major players that affect gene expression are the cAMP-CRP complex and RNA polymerase. As mentioned above, $k$ for direct binding genes reflects the affinity of the cAMP-CRP complex to that of the promoter, while $E_{max}$ is affected by promoter properties such as RNAP binding and CRP binding to the promoter. Because $k$ and $E_{max}$ may capture the physiological effects of cAMP-CRP and

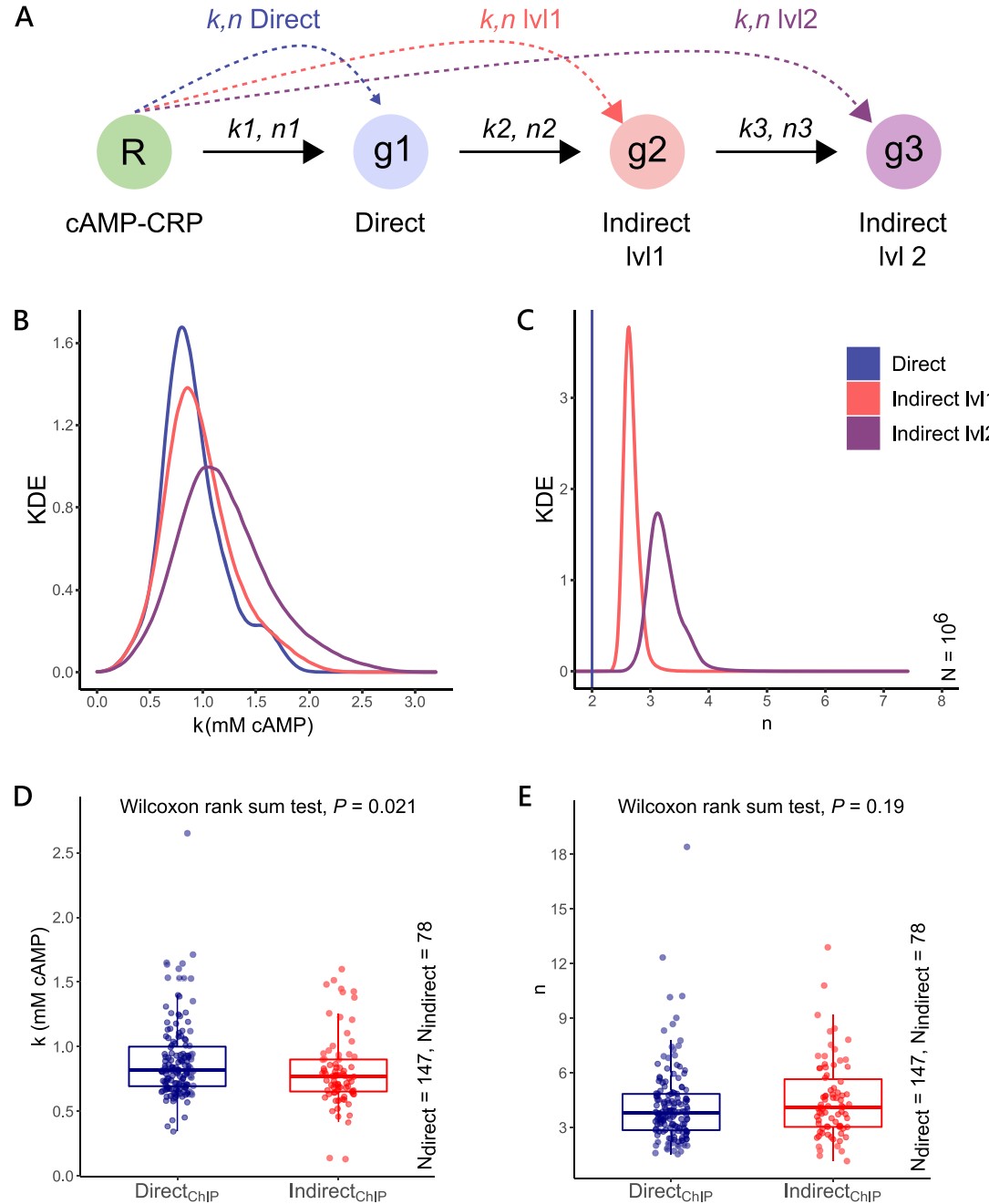

**FIG 5** Comparison of *k* and *n* across direct and indirect genes. (A) Toy model of a linear transcription network representing a regulator (R), a directly regulated gene (g1), and two indirectly regulated genes (g2 and g3). Each edge follows the Hill's function. *k* at each edge is independent of other edges and is chosen at random from the same distribution. $E_{max}$ and *n* remain constant across the circuit, with $n1 = n2 = n3 = 2$, $b0 = 0$, and $E_{max} = 1$. We calculate the apparent *k* and *n* of the response for g1 (direct), g2 (indirect lvl1), and g3 (indirect lvl2) with respect to the regulator (R), represented by the dotted lines. (B and C) Expected distribution of apparent *k* and *n* at each level of the regulatory network compared to the *k* and *n* for direct genes ($10^6$ trials). KDE represents Kernel Density Estimation. According to this null model, the response of genes further down the regulatory circuit show greater mean and variance for both *k* and *n*. (D and E) Observed distributions of *k* and *n* across Direct$_{ChIP}$ (147) and Indirect$_{ChIP}$ (78) sets of cAMP-regulated genes. Contrary to the null model, distributions of *k* and *n* across direct and indirect genes show no difference in their mean and variance. We used the Wilcoxon rank-sum test to compare the two distributions.

RNAP binding to the promoter on gene expression, we asked if their binding strengths have any effect on the variation of these parameters across cAMP-regulated genes. We used data from ChIP experiments in *E. coli* as a proxy for binding affinities of CRP and RNAP. We assumed the strength of the signal at a position to be proportional to the

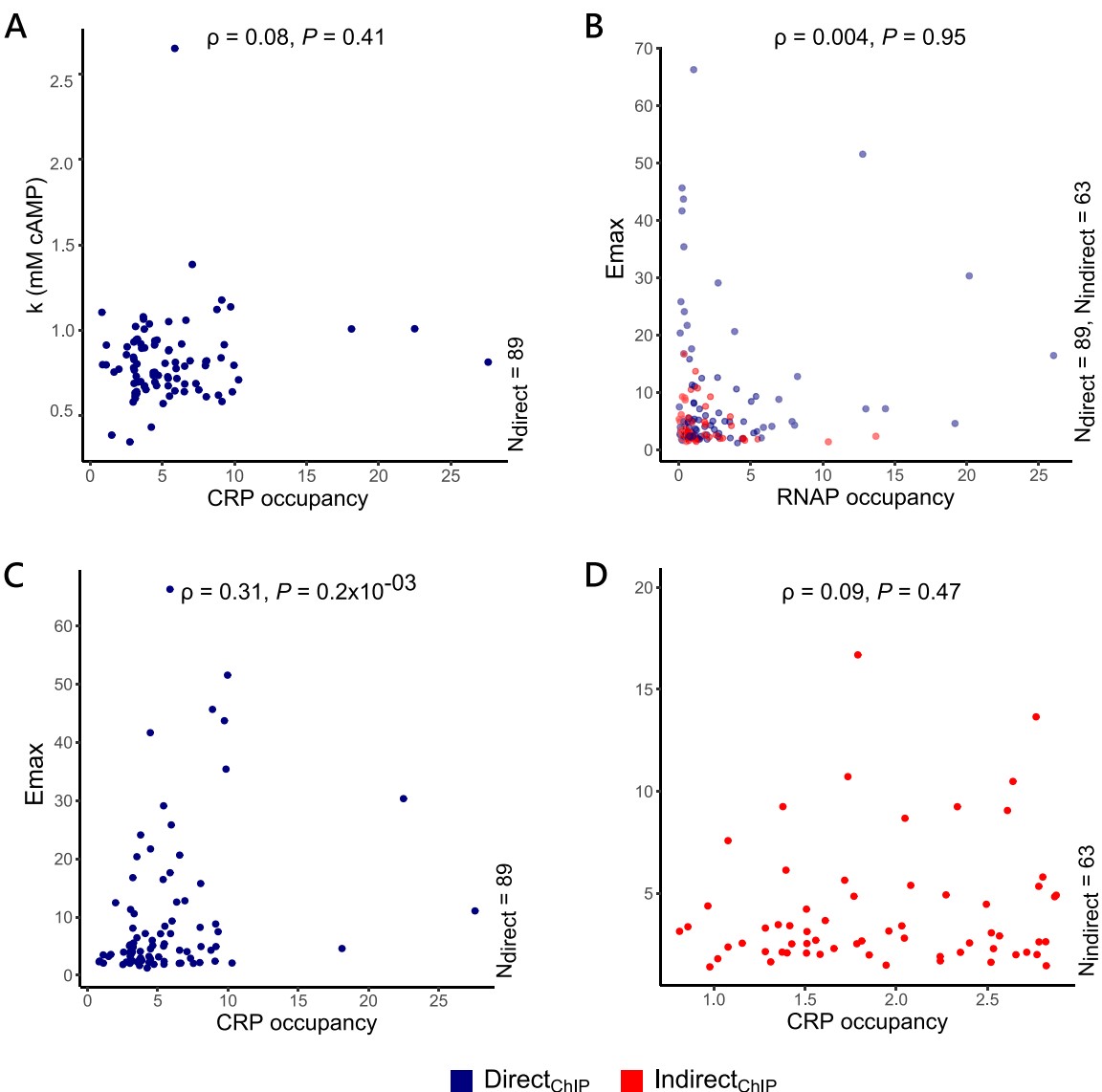

**FIG 6** Effects of CRP and RNAP occupancy on $k$ and $E_{max}$ of cAMP-regulated genes. (A) Correlation between $k$ and CRP occupancy at promoters for genes under direct regulation of the cAMP-CRP complex (Direct$_{ChIP}$, 89 promoters). (B) Correlation between $E_{max}$ and RNAP occupancy for both Direct$_{ChIP}$ (89 promoters) and Indirect$_{ChIP}$ (63 promoters). $E_{max}$ and RNAP occupancy for either set of genes are not correlated. (C and D) Correlation between $E_{max}$ and CRP occupancy for Direct$_{ChIP}$ (C) and Indirect$_{ChIP}$ genes (D). $E_{max}$ of genes under direct regulation of cAMP show a significant correlation with CRP occupancy at their promoters. $\rho$ and $P$ in the plots represent the Spearman's rank correlation coefficient and the corresponding $P$ value for the given pair of variables.

probability of the target molecule occupying that position, which in turn is affected by their effective binding affinity.

We checked the correlation between $k$ and cAMP-CRP occupancy scores for Direct$_{ChIP}$ genes. $k$ of cAMP-regulated genes did not show any correlation with cAMP-CRP occupancy (Fig. 6A; Fig. S8A and B), indicating that CRP occupancy alone was not sufficient to explain the observed values of $k$. These results hinted that for a cell growing in a specific medium, individual binding affinities may not play a large role in determining the midpoint of the dynamic range of genes.

$E_{max}$ of a gene reflects the maximum change in expression (fold change) it can experience when cAMP is not limiting, that is, [cAMP] $\gg k$ and reflects the maximum sensitivity of a promoter to cAMP. $E_{max}$ is affected by promoter properties, such as RNAP binding affinity, interaction of CRP, other regulators, and different NAPs with the RNAP, promoter escape rates, and gene dosage (66–68). We observed no correlation between $E_{max}$ of

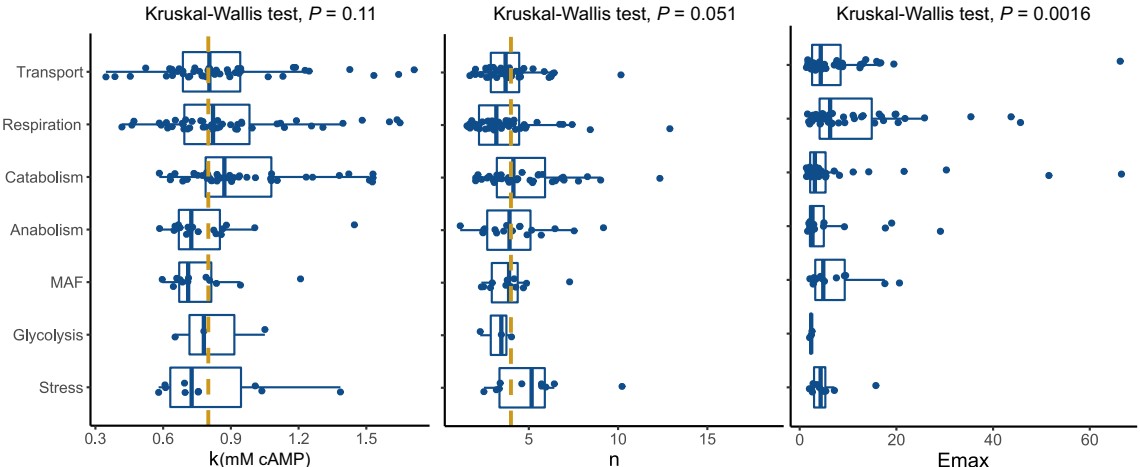

**FIG 7** Distribution of $k$, $n$, and $E_{max}$ for genes belonging to various functional categories. There are no significant differences in $k$ and $n$ across all metabolic pathways. $E_{max}$ values of genes involved in respiration are significantly higher than those involved in catabolism and anabolism (Fisher's test, adjusted $P < 0.05$). Dashed lines represent the median of the distributions.

cAMP-regulated genes with RNAP occupancy at the promoter (Spearman's rank correlation coefficient, $\rho = 0.004$, $P = 0.95$; Fig. 6B). CRP occupancy, on the other hand, showed a small but significant correlation of 0.37 (Spearman rank correlation, $P = 2.14 \times 10^{-6}$) with $E_{max}$ (Fig. 6C and D; Fig. S8C and D). We observed that genes belonging to Direct$_{ChIP}$ had a significant correlation between $E_{max}$ and CRP occupancy (Spearman rank correlation coefficient, $\rho = 0.31$, $P = 0.2 \times 10^{-3}$; Fig. 6C), while Indirect$_{ChIP}$ genes did not show any correlations (Spearman rank correlation coefficient, $\rho = 0.09$, $P = 0.47$; Fig. 6D), showing that the observed correlation between $E_{max}$ and CRP occupancy for cAMP-activated genes was driven by direct-binding genes.

In this section, we have attempted to quantify the individual effects of two major molecular players, CRP and RNAP, on the dynamic range ($k$) and sensitivity ($E_{max}$) of genes under direct control of cAMP. Our data showed that despite having different binding affinities to the promoters, under physiological conditions, cAMP-CRP binding affinities have no effect on the concentration of cAMP required for the gene expression to reach its half-saturating concentration. We also showed that the sensitivity of genes is correlated with a small but significant degree to the differential binding of CRP at promoters but not RNAP.

**Variation in phenomenological constants across various functional groups.** CRP is well known for its role in regulating genes involved in the uptake and utilization of multiple carbon sources. Apart from this, the cAMP-CRP complex has also been shown to regulate genes involved in nitrogen metabolism, the tricarboxylic acid (TCA) cycle, osmoregulation, and antibiotic resistance (34, 40). We binned the genes positively regulated by cAMP into broad functional categories (69, 70). Of these, we focused on seven broad categories, namely, catabolism, anabolism, glycolysis, mixed acid fermentation (MAF), respiration, transport, and stress response (Fig. S9A and B). cAMP-regulated genes were enriched for carbon catabolism, transport, and respiration (odds ratio [OR] > 1, $P < 0.05$) (71–73). We found that genes belonging to HM, NR, and NM were not enriched for any particular functional group. However, LM genes showed an enrichment for respiration genes (OR > 4.3, $P = 0.0052$). We also noted that 11/13 genes in the LM that were enriched for respiration belong to only four operons (*ccm*, *fdn*, *nar*, and *nir*). Because most cAMP-regulated genes belonged to HM (225/305), we checked if genes belonging to different functional categories behave differently. We compared the $k$, $n$, and $E_{max}$ of genes across these functional categories (Fig. 7) and found no difference in $k$ or $n$ of genes belonging to any category. However, genes involved in respiration showed significantly higher $E_{max}$ than carbon catabolism genes (Wilcoxon rank-sum test, corrected $P < 0.01$).

Transport and catabolism mainly included genes involved in uptake and utilization of various carbon compounds and together accounted for 35% of the differentially expressed genes. Most genes in this category were involved in the uptake and breakdown of carbohydrates (67), glycerol (10), and amino acids (13). *E. coli* is able to catabolize a wide range of carbohydrates, such as simple and complex sugars, organic acids, and polyols. In line with this, we observed an upregulation of genes such as *fruBKA*, *malE*, *malK*, *lamB*, *galP*, *manXYZ*, *gatYZABCD*, *mtlAD*, *uxaB*, *uxaC*, *rbsDACBKR*, *malP*, and *malQ*. $k$, $n$, and $E_{max}$ of carbohydrate genes spanned the entire biological range of cAMP (Fig. S9C and D). Utilization of a carbohydrate involves both its uptake via transporters and breakdown by catabolic enzymes. Many specific and nonspecific transporters were activated by cAMP in LB. While the distributions of $k$ and $E_{max}$ for catabolic and transport genes do not differ significantly, genes encoding carbohydrate catabolism proteins show significantly higher $n$ than genes involved in transport (Fig. 8A and B). This suggests that genes involved in catabolic roles may be under more complex control and behave in a more switch-like manner than genes encoding transporters which behave in a relatively more graded manner.

*E. coli* can also utilize other molecules as a source of carbon. We observed genes for uptake and utilization of small peptides and amino acids (*pepE*, *pepT*, *sdaC*, *tdcB*, and *tnaA*), nucleic acid derivatives such as cytidine and uridine (*cdd*, *deoA*, and *deoC*), and glycerol and phospholipids (*glpABC* and *glpQT*) to express after addition of cAMP. For this study, we considered values of $n$ greater than 5 to be high, values of $n$ between 3 and 5 to be moderate, and values of $n$ less than 3 to be low. For $k$, we considered values of $k$ less than 0.68 mM cAMP to be low, between 0.68 and 0.94 mM cAMP to be moderate, and greater than 0.94 mM cAMP to be high. We found that most amino acid catabolism genes had low $k$ and high $n$, followed by nucleic acid genes that showed moderate values for both $k$ and $n$, close to the population median. Glycerol and lipid catabolism genes showed moderate to high values of $k$ with moderate to low values of $n$ (Fig. 8C and D). This meant that expression of amino acid genes activated at low cAMP concentrations in a switch-like manner, unlike nucleic acid and glycerol catabolism genes whose response increased in a relatively graded fashion in response to cAMP changes in the cell.

Genes of upper glycolysis (*pfkA*, *pykA*, and *gpmA*) showed an upregulation in response to cAMP. We also found MAF genes (*pflB*, *adhE*, *pta*, and *ackA*) to be upregulated in response to cAMP. It is not uncommon for *E. coli* to opt for such overflow metabolism during rapid phases of growth (73, 74). Expression of MAF and glycolytic genes was limited to low and mid ranges of $k$ and moderate values of $n$, close to the population median.

We observed an upregulation of respiration genes as well on addition of cAMP. The respiration genes exhibited a significantly higher $E_{max}$ than catabolic genes. Genes involved in respiration spanned the whole range of $k$ and $n$ and showed very high expression ($E_{max}$) in response to cAMP. Respiration consisted of genes involved in both aerobic and anaerobic respiration and the electron transport chain (ETC) (Fig. S9E and F). Very few aerobic respiration (*cydA* and *ccmA*) and ETC (*ndh*) genes were differentially expressed in LB in response to cAMP. In *E. coli*, *cydA* and *ccmA* encode cytochromes. These genes responded to low cAMP concentrations in a switch-like manner and achieved moderate values of $E_{max}$, with an ~5-fold change. Few anaerobic respiration genes on the other hand exhibited very high $E_{max}$ and spanned across the range of $n$ and $k$. These sets of genes included *narGHJI*, *napFDAGHBC*, and *dmsBC*. We checked the occupancy of CRP at these promoters and found a strong correlation between $E_{max}$ and CRP occupancy of these genes (Pearson correlation, $r = 0.89$, $P = 1 \times 10^{-6}$; Fig. 8E). This could explain why genes related to anaerobic respiration express at such high $E_{max}$ despite the cells growing under aerobic conditions.

## DISCUSSION

In this paper we showed the use of phenomenological models like the Hill's model as a tool to gain insights about the cAMP regulatory network. For this study, we generated genome-wide gene expression profiles of *E. coli* as it responds to 10 different

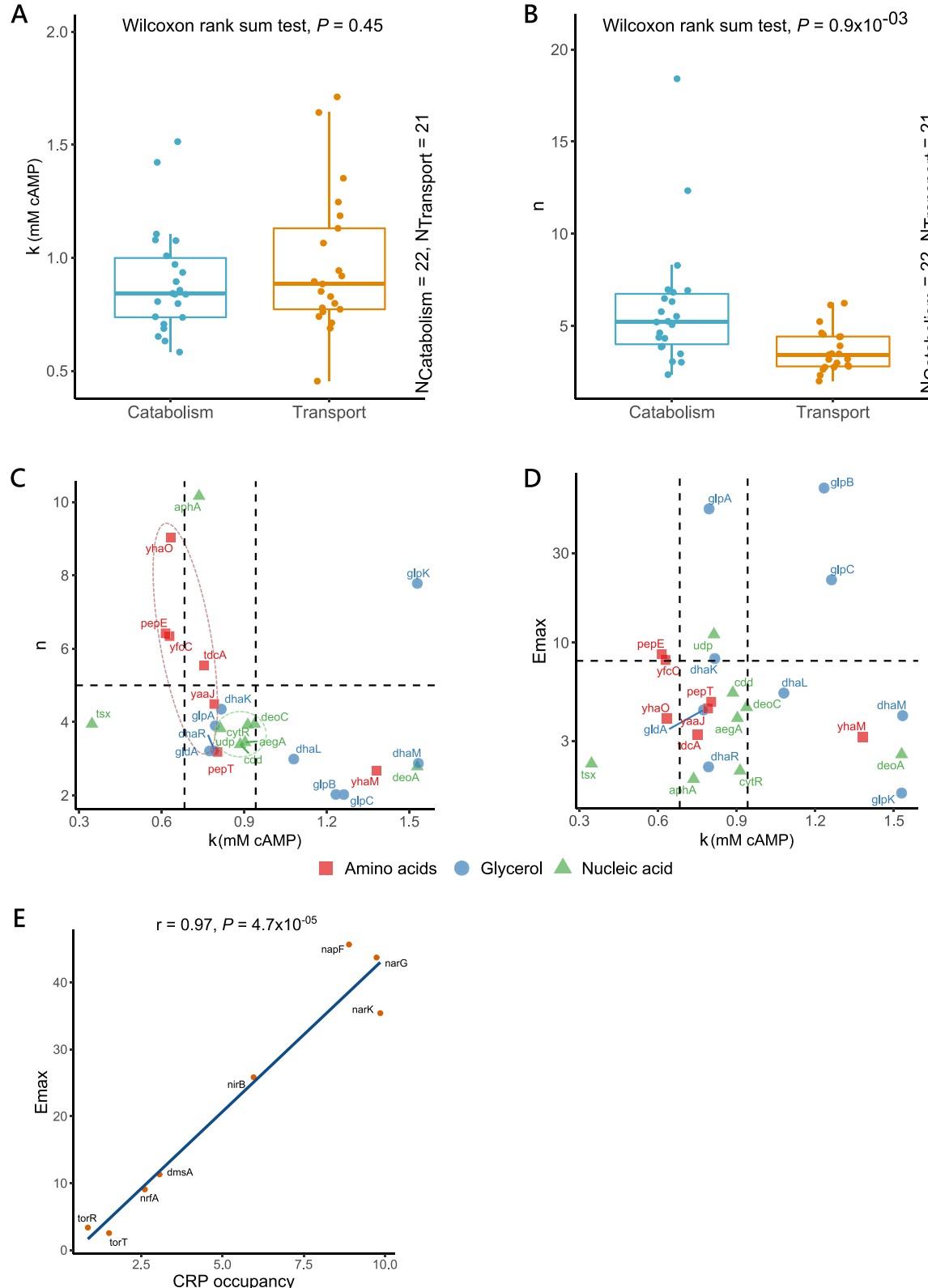

**FIG 8** Comparison of $k$, $n$, and $E_{max}$ across metabolic classes. (A and B) Distributions of $k$ and $n$ for carbohydrate catabolism (22) and transport (21) genes. Median of distribution of $n$ for catabolic genes is higher than those of transporter genes. (C and D) Scatterplot showing the relationship between $n$ and $k$ (C) or $E_{max}$ and $k$ (D) for genes involved in catabolism of amino acids, glycerol, and nucleic acid. Dashed lines reflect the top quartile (Q3) for $n$ and the IQR for $k$. Amino acid catabolism genes show higher $n$ and lower $k$ than genes encoding glycerol and nucleic acid catabolism. (E) Scatterplot showing the correlation between $E_{max}$ and CRP occupancy for anaerobic respiration genes. $r$ and $P$ represent the Pearson correlation coefficient and the associated $P$ value for the pair of variables.

concentrations of cAMP. Our data revealed that the response of gene expression to cAMP can be nonmonotonic, sigmoid, or linear or have more complicated responses that are intermediate superpositions of these simple shapes, thus making the space of possible trends to be populated continuously and without distinct clusters. To increase the interpretability of such a data set, we summarized the response of individual genes using phenomenological models that best capture the behavior of their responses. Further, we explored the use of phenomenological constants obtained from the models to understand properties of the cAMP regulatory network. A majority (74%) of the cAMP-regulated genes followed a sigmoid-shaped curve, which could successfully be described by the Hill's model.

Our data suggested that promoters and intracellular cAMP concentrations in the cAMP regulatory network may be tuned, resulting in optimum gene expression. Most of cAMP-regulated genes showed switch-like behavior, with the midpoints of their dynamic range centered around the wild-type concentrations of cAMP (0.72 mM cAMP), resulting in a rapid burst of gene expression necessary to reach wild-type growth rates in LB as soon as the wild-type concentration is reached. Even after adding concentrations of cAMP much greater (2 to 4 times) than those found in wild-type cells in LB, the excess cAMP was not able to induce non-LB-specific genes. Combinatorial control of environmental and transcription factors for gene expression is well known. As cAMP concentrations increase, the cell concurrently unlocks and primes increasing numbers of metabolic modules. However, environmental signals and metabolic feedback tightly control which of these primed modules will actually be expressed in a condition-specific manner (17, 19, 21, 27, 28, 75, 76).

Expression of cAMP-regulated genes differed largely in their $E_{max}$ compared to $k$. We found that CRP occupancy explained the variation in $E_{max}$ to a small but significant degree but showed no correlation to $k$. However, we found gene expression to be independent of the RNAP occupancy at gene promoters. Together these observations imply that (i) a high-affinity CRP promoter need not ensure transcriptional activation at low concentrations of cAMP, but instead, it is more likely to control the magnitude of a gene's response to cAMP concentrations, and (ii) promoter properties other than RNAP and CRP binding alone play a role in determining the levels of gene expression in response to cAMP. Predictive models for transcriptional regulation as well as some experimental data have shown the contributions of factors such as RNAP-CRP synergy, promoter escape rates, gene dosage, and the presence of secondary coactivators in determining the $E_{max}$ (66–68, 75, 77).

It is well known that feedforward and feedback loops are pervasive in the *E. coli* transcriptional network (49, 62, 78). The presence of high values of $n$ for cAMP-regulated genes confirmed that. The cAMP regulon showed a lack of genes that respond in a highly switch-like manner ($n$) at high concentrations of cAMP ($k$). This trend remained consistent across both direct and indirect genes (Fig. S6D). Based on our null model of a linear regulatory circuit, we expected $n$ and $k$ of indirect genes to have greater nonlinearity and higher variation in their dynamic range. However, the observed data showed no differences in the distribution of either $k$ or $n$ across these sets of genes. We suspect that a combination of short length subnetworks with extensive feedforward and feedback loops feeding into higher nodes could be the reason leading to higher variation in distributions of $k$ and $n$. These expectations are based on the fact that a large fraction of cAMP-regulated genes are involved in carbon catabolism, which have previously been shown to have short transcriptional circuits with multiple feed-forward loops (78–80).

In the final section of the paper, we studied how $k$, $n$, and $E_{max}$ are tuned for various metabolic pathways in *E. coli* growing in LB. The most enriched class of genes was carbon catabolism and transport. $k$, $n$, and $E_{max}$ for carbohydrate catabolism and transport genes spanned across the whole measured range. Because genes involved in uptake of different carbohydrates are made up of independent modules, these genes show a broader variation in their $n$ and $k$. Contrary to carbohydrates, genes involved in amino acid, nucleic acid, and glycerol degradation, which included genes participating in a single pathway, showed more clustered distributions for $n$ and $k$ (78, 80). However, the numbers of participating genes were too few to do any statistical analysis on these trends.

We understand that phenomenological models do not have predictive powers and can at best be used as a descriptive tool. Despite the large range of cAMP used in this study, predictions of parameters such as $n$ and $b0$ remained relatively poor owing to lack of resolution at transition points and errors induced by small measures. While it is important to understand the response of genes at a physiological level, this method allowed us to study the response of genes only relative to cAMP and shed very little light on the intricate deeper levels of regulation between individual players in the regulatory circuit. Another caveat is that our study is blind to the composition of ingredients in LB. Thus, for genes that have two or more input functions (inducible genes), this approach quantitates only the physiological $k$, $n$, and $E_{max}$. This makes the values of these phenomenological constants limited to one condition. However, with the advent of less expensive, faster, and deeper RNA-seq technologies, such studies can be extended to other carbon sources in a controlled environment.

We find phenomenological models useful to quantitatively describe dose-response curves for transcription factors at a global level. This method also circumvents the problems and limitations posed by conventional clustering techniques. Further, the interpretations of the Hill's parameters can be extended to global regulatory networks to understand its topology. We hope this kind of an approach can provide essential raw material required to ask broader mechanistic questions about transcriptional regulation in bacteria.

## MATERIALS AND METHODS

**Growth conditions and intracellular cAMP measurements.** *Escherichia coli* K-12 MG1655 was used as the wild-type strain for this study. The Δ*cyaA* strain was obtained from A. Mogre (36). The *E. coli* mutant was constructed by deleting the *cyaA* gene from the wild-type strain using methods described by Datsenko and Wanner (81). Cells were grown in either Luria-Bertani broth (LB; Hi-Media, M575-500) or M9 minimal medium with 0.4% sugars. M9 salts (12.8 g/L Na$_2$HPO$_4$•7H$_2$O, 3 g/L KH$_2$PO$_4$, 0.5 g/L NaCl, and 1 g/L NH$_4$Cl) were supplemented with 2 mM MgSO$_4$, CaCl$_2$, 1 mM thiamine, 0.2% casamino acids, and 0.4% (wt/vol) carbon source (lactose or sorbitol-ribose mixture).

To study the effects of various cAMP concentrations on growth kinetics and the transcriptome of Δ*cyaA E. coli* cells, cAMP sodium salt (adenosine 3′,5′-cyclic monophosphate sodium salt monohydrate; Sigma-Aldrich, A6885) was added to the growth medium, obtaining 10 different final concentrations ranging from 0 mM to 4 mM (22). Wild-type *E. coli* cells were used as the control in this experiment. Overnight-grown cultures were inoculated at a 1:100 dilution in 50-mL flasks containing 15 mL of fresh medium with various amounts of cAMP. Cultures were grown at 37°C at 180 rpm. OD measurements were taken at 600 nm every 30 min for the first 4 h and every 1 h after that. Maximum growth rate ($\mu_{max}$) of a population was calculated using the growthcurveR package in R (82). The time the bacterial culture took to reach its $\mu_{max}$ was considered to be the "lag time."

Intracellular cAMP concentrations corresponding to the extracellular cAMP were measured using cAMP enzyme-linked immunosorbent assay (ELISA) kits provided by Cayman Chemicals. Δ*cyaA* cells were grown in different cAMP concentrations ranging from 0.01 mM to 2 mM, and cells were harvested when the wild-type population growth reached their $\mu_{max}$. Harvested cells were washed using 1× phosphate-buffered saline (PBS; 8 g/L NaCl, 0.2 g/L KCl, 1.44 g/L Na$_2$HPO$_4$, and 0.24 g/L KH$_2$PO$_4$, pH 7.4) to remove any remaining medium and cAMP. Cells were resuspended in 1× PBS and boiled at 95°C for 10 min. Cell debris was removed by centrifugation at >10,000 rpm (36). The supernatant was analyzed using the protocol provided by the cAMP ELISA kits. We found that intracellular cAMP increases linearly with addition of extracellular cAMP (Fig. S1G in the supplemental material). As a control, intracellular cAMP concentrations of wild-type and Δ*cyaA* mutants were also measured (Fig. S1B).

**DNA isolation and sequencing.** Genetic backgrounds of wild-type and Δ*cyaA*-mutant cells were confirmed by whole-genome sequencing. Overnight-grown cultures were used to inoculate 50 mL of fresh LB medium to make a final dilution of 1:100. Cells were grown at 180 rpm at 37°C and harvested at their $\mu_{max}$. Genomic DNA was isolated using a GenElute bacterial genomic DNA kit (NA2120-1KT, Sigma-Aldrich) using the manufacturer's protocol. Library preparation was done using Truseq Nano DNA library preparation kits followed by paired-end (2 × 100) sequencing using Illumina HiSeq 2500 platform. Genome sequences were analyzed for single-nucleotide polymorphisms (SNPs) and insertions/deletions (indels) using the breseq software and protocol described by the Barrick lab (83). Apart from the expected loss of the *cyaA* gene in the Δ*cyaA* mutant, we found a 1,063-bp deletion at position 1977440 in the *flhC* gene of the *flhDC* operon. This deletion is absent in the wild-type strain. This operon is a known target of the cAMP-CRP signaling system. RNA-sequencing results show that the *flhDC* operon and downstream gene *fliA* encoding $\sigma^F$ fail to respond to high doses of extracellular cAMP, resulting in shutting down of flagellar-, motility-, chemotaxis-, and biofilm-related genes in the Δ*cyaA* mutant (84, 85).

**RNA sequencing and differential expression analysis.** Δ*cyaA* cells grown in LB with different cAMP concentrations were harvested when the wild-type population reached $\mu_{max}$. Two replicates for each sample were processed. Total RNA was isolated using the TRIzol-chloroform extraction method followed by DNase treatment. The 16S and 23S rRNA were depleted using Ambion MICROBExpress bacterial mRNA enrichment kits (AM1905), and the RNA quality was checked using a Bioanalyzer (Agilent) followed by

Qubit quantification. Libraries for each sample were prepared using the New England Biolabs (NEB) NextUltra directional RNA library prep kit followed by single-end sequencing using the Illumina HiSeq 2500 platform.

All annotation and sequence files were obtained from NCBI. *E. coli K*-12 MG1655 (NC_000913.2) was used as the reference genome for RNA sequence analysis. Sequencing reads were aligned and mapped to the reference genome using the Burrows-Wheeler aligner (BWA) algorithm. SAMtools (v1.2), and BEDtools (v2.25.0) were used to determine read counts per gene. Normalization and differential gene expression analysis across samples was done using the EdgeR package (3.28.1) in R as described by Chen et al. (86).

For any pairwise comparison, differentially expressed genes were defined as the set of genes showing a $\log_2$ (Fold Change) of $\pm 1$ and a *P* value of $<0.01$. Genes differentially expressed between the wild-type strain and the $\Delta cyaA$ mutant were considered to be cAMP responsive. The effect of any cAMP concentration on a particular gene was quantified as the fold change a gene experienced at that cAMP concentration compared to the 0 mM cAMP $\Delta cyaA$ state. Since log scale is nonlinear and will affect the shape and magnitude of individual trends, all $\log_2$ (Fold Change) values produced by EdgeR were converted to fold change before further analysis.

We find that genes encoding proteins involved in flagella, chemotaxis, and biofilm formation, which are under the control of *flhDC* and $\sigma^F$, do not respond to even high concentrations of extracellular cAMP. For this study, we have removed them from the set of cAMP-responsive genes (Fig. S2A).

**Chromatin immunoprecipitation sequencing (ChIP-seq) for CRP.** The ChIP method was adapted from a previous study with few changes (59). Cells were grown aerobically at 37°C to the early exponential phase ($\sim$OD of 0.2). Formaldehyde was added at a final concentration of 1% and incubated for 20 min at 37°C. Glycine was added to quench the cross-linking at a final concentration of 0.5 M, and cells were incubated for 5 min. Cells were then harvested and washed three times with cold 1$\times$ Tris-buffered saline (TBS) and resuspended in 1 mL of lysis buffer (10 mM Tris [pH 8.0], 20% sucrose, 50 mM NaCl, 10 mM EDTA, 20 mg/mL lysozyme, and 0.1 mg/mL RNase A) and incubated at 37°C for 30 min. After the incubation, 3 mL of immunoprecipitation buffer (IP buffer; 50 mM HEPES-KOH [pH 7.5], 150 mM NaCl, 1 mM EDTA, 1% Triton X-100, 0.1% sodium deoxycholate, 0.1% sodium dodecyl sulfate [SDS], and phenylmethylsulfonyl fluoride [PMSF; final concentration of 1 mM]) was added. Cells were then sheared in a Bioruptor (Diagenode) with 33 cycles (25 s on/24 s off). Cellular debris was removed by centrifugation at 4°C, and the supernatant was split into aliquots for ChIP and input samples. Each aliquot was incubated with 1$\times$ TBS pre-washed 20 $\mu$L of protein A/G ultra-link resin beads on a rotary shaker for 45 min at room temperature. Samples were then centrifuged, and the supernatant was incubated with flag mouse monoclonal antibody (IP; Sigma-Aldrich) and no antibody (mock IP) for 60 min at room temperature. Meanwhile, 40 $\mu$L of A/G ultralink resin beads (Thermo Fisher Scientific) per sample was blocked in 1 mg/mL bovine serum albumin (BSA). After 1 h of incubation, 40 $\mu$L of blocked A/G ultralink resin beads was added to all samples and incubated for 90 min at room temperature. Beads were then collected after centrifugation in SpinX-Costar tubes. After collection, beads were then washed successively in IP buffer, twice with high-salt IP buffer (IP buffer + 500 mM NaCl), once with wash buffer (10 mM Tris [pH 8.0], 250 mM LiCl, 1 mM EDTA, 0.5% Nonidet P-40, and 0.5% sodium deoxycholate), and once with Tris-EDTA, pH 7.5 (TE). All samples were washed rotating the tubes in a rotary shaker for 3 min and centrifuging at 3,500 rpm for 2 min. Immunoprecipitated complexes were eluted in 100 $\mu$L of elution buffer (10 mM Tris [pH 7.5], 10 mM EDTA, and 1% SDS) at 65°C for 20 min. After elution, the sample along with the input was reverse cross-linked in 0.5$\times$ elution buffer + 0.8 mg/mL Pronase (Sigma-Aldrich) at 42°C for 2 h followed by 65°C for 6 h. DNA was then purified using a MinElute PCR purification kit (Qiagen). After ChIP experiments, 5 ng of DNA from antibody-treated samples and input was taken to prepare libraries using an Illumina TruSeq DNA preparation kit. All the steps were performed using manufacturer's instructions, followed by paired-end sequencing. The paired-end reads after adaptor trimming were aligned to the *E. coli* reference genome (NC_000913.2) using BWA. Reads per base were calculated using SAMtools (87).

**Estimating CRP and RNAP occupancy at gene promoters.** *In vivo* CRP and RNAP occupancy was calculated from CRP and RNAP ChIP-seq experiments performed in *E. coli* cells at an exponential phase in LB. The occupancy at any genomic region was considered to be proportional to the intensity of the ChIP signal at that base pair. Occupancy, in turn, is used as a proxy for the affinity of CRP and RNAP for that region.

To calculate occupancy at specific genomic regions from ChIP-seq data, reads per base pair were obtained. It was ensured that the reference genome used for alignment in ChIP studies corresponded to the one used for RNA-sequencing experiments (NC_000913.2). For both data sets, reads per base were internally normalized by dividing the frequency at each base by the mode of the distribution, and the signal at each base was calculated as the ratio of the test signal to that of the internal control. Spurious peaks were removed by smoothing the data using a local regression method. The maximum frequency recorded within a given region was considered the occupancy for that region. A region of $-200$ to $+50$ bp flanking the start site of the gene was considered the CRP- or RNAP-binding region. For genes that are part of operons, the occupancy score for only the first gene was considered for all analyses.

CRP affects gene expression by recruitment of RNAP at the promoter. Despite being from two different studies, a significant correlation between CRP and RNAP occupancy (Pearson correlation coefficient, $r = 0.36$, $P = 7.7 \times 10^{-6}$; Fig. S8E) was observed, showing that the data sets are internally consistent.

**Model fitting and estimation of parameters.** All dose-response curves were calculated as the fold change of a gene at a given cAMP concentration compared to the $\Delta cyaA$ mutant. All analysis was done using custom scripts in R. All curves were fit to different models using the nls function from the R stats package.

**(i) Model definition.** A gene is expressed when the cAMP-CRP active complex binds to the

promoter of a gene. Based on the underlying regulatory mechanisms, gene expression versus cAMP concentration response curves may follow one of the following trends: no response, linear/nonsaturating, or sigmoid. We checked the fit of each gene to these three models.

**Hill's model (HM).** Nonlinear relationships, especially sigmoidal ones, are common in transcriptional control of gene expression by activators and inhibitors (60). Gene expression following a sigmoidal response curve can be captured well using phenomenological models like the four-parameter Hill's model, defined by $b0$, $n$, $k$, and $E_{max}$ (Fig. 4A). Hill's model can be derived by considering the equilibrium binding of a transcription factor to its promoter (51, 53, 57, 88, 89). We held the following assumptions while applying the Hill's model to our data: (i) the extracellular cAMP immediately equilibrates with the intracellular cAMP, (ii) the active cAMP-CRP complex is proportional to extracellular cAMP, and (iii) RNAP is a not a limiting factor for transcription (90). For an activator, such as the cAMP-CRP complex, the Hill's curve can be written as

$$\text{Expression[cAMP]} = b0 + (E_{max} - b0) \cdot [\text{cAMP}]^n / (k^n + [\text{cAMP}]^n) \tag{1}$$

The biological relevance of the phenomenological constants is well hashed out in the field (50, 56). We extend these definitions to a network:

(i) $b0$ is the baseline expression of a gene in the cell when cAMP is absent in the system. Biologically, this can be interpreted as the leaky expression of the system and depends on how tightly repressed a gene is. In our study, $b0$ is ~1 fold change.

(ii) $E_{max}$ represents the expression level when the concentration of the cAMP-CRP complex has far exceeded its binding sites, and transcription is no longer limited by the concentration of intracellular cAMP. Physically, it is affected by intrinsic promoter properties, such as RNA polymerase (RNAP) binding strength to the promoter, the interaction of cAMP-CRP or other NAPs binding at the promoter, interaction between the cAMP-CRP complex and RNAP, effects of gene dosage, and promoter escape rates on levels of transcription (66, 67, 77, 91). Biologically, $E_{max}$ tells us the maximum sensitivity of a gene in response to cAMP. $E_{max}$ is measured in terms of fold change compared to the $\Delta cyaA$ mutant and ranges from 1 to infinity.

(iii) $k$ represents the cAMP concentration at which half the saturating expression has been achieved. Biologically, it is the midpoint of the dynamic range of the gene. For switch-like genes, it reflects the cAMP concentration at which the gene starts expressing. Physically, for genes that are regulated by the direct binding of the cAMP-CRP complex to their promoters, $k$ reflects the affinity of the complex to the promoter. For genes that are under indirect regulation of cAMP, $k$ reflects the composite effects of all $k$ values in the network (57).

(iv) $n$ determines the extent of graded versus switch-type response the gene has upon activation. For a single promoter-cAMP-CRP complex pair, this reflects the cooperative behavior of the transcription factor. It implies, either enhanced binding or decreased unbinding of the transcription factor as a function of cAMP concentration. For gene regulatory networks, $n$ indicates the presence of positive feedback and multistep feedforward loops (60, 61, 89). The steeper the response curve, the higher the $n$.

**Linear/nonsaturating model (LM).** Genes whose saturating concentrations are well beyond the cAMP concentrations administered in our study will fail to show saturation under our cAMP regimen. The Hill's equation from the previous section can be modified to a nonsaturating model:
for $k \gg [\text{cAMP}]$,

$$\text{Expression[cAMP]} = b0 + (E_{max} - b0) \cdot [\text{cAMP}]^n / k^n.$$

Since $(E_{max} - b0)/k^n$ will be a constant for each given curve, the above equation can be reduced to

$$\text{Expression[cAMP]} = b0 + m \cdot [\text{cAMP}]^n. \tag{2}$$

A linear relationship between gene expression and cAMP concentration is a special case of equation 2 where $n = 1$. This can mathematically be represented by

$$\text{Expression[cAMP]} = b0 + m \cdot [\text{cAMP}], \tag{3}$$

where $m$ is the slope of the line, and $b0$ is the basal expression of the gene.

**Nonresponsive model (NR).** The null model for gene expression response to cAMP is given by a no expression model.
For $[\text{cAMP}] = 0$

$$\text{Expression[cAMP]} = b0. \tag{4}$$

**(ii) Goodness of fit measurements.** The best fit for each model was estimated using methods that minimize the sum of squared estimate of errors (SSE), where the SSE is defined by

$$\text{SSE} = \sum_{i=1}^{N} (y_i - f(x_i))^2,$$

where for $N$ number of observations, $y_i$ is the $i^{th}$ value of the variable to be predicted, $x_i$ is the $i^{th}$ value of the explanatory variable, and $f(x_i)$ is the predicted value of $y_i$.

A model with lower estimates of SSE was considered to be the better fit. SSE is a meaningful measure when comparing competing models. However, it does not reflect on how well the fitted model explains the observed data. For a linear regression, the Pearson's $R^2$ coefficient of determination gives a statistical measure of how well the regression predictions explain the observed data. $R^2$ is measured as

$$R^2 = 1 - \frac{SSE}{TSS},$$

where TSS is the total sum of squares and is given by

$$TSS = \sum_{i=1}^{N} (y_i - \bar{y})^2,$$

where $y_i$ is the $i^{th}$ value in the sample, and $\bar{y}$ is the mean of the sample.

However, $R^2$ is not calculated for nonlinear curve fits and has been shown to give inconsistent results. Here, we try to define a similar metric for the nonlinear regression model. To do so, we calculated a linear regression between the values predicted using the fitted function and the observed expression data for each cAMP concentration. The Pearson $R^2$ value obtained ($R^2_{NLfit}$) quantifies how well the variation in the observed data is predicted by the model. Along with $R^2_{NLfit}$ values, the slope between the observed data and predicted data was also calculated (slope$_{NLfit}$). The associated slope quantifies how close the predicted values are to the observed data.

Finally, to determine the accuracy of each estimated parameter ($b0$, $n$, $k$, $E_{max}$, or $m$), a relative standard error (RSE) was calculated by

$$RSE = \frac{\text{Standard error of the estimate}}{\text{Predicted estimate value}}.$$

**(iii) Model fitting and parameter estimation.** The nls function from the R Stats package was used to fit the three models (no response, nonsaturating/linear, and Hill) to each gene curve.

The nls function returns the residual standard error ($\sigma^2$) value for the best possible fit given each model,

$$\sigma^2 = \frac{SSE}{N - p},$$

where $N$ is the number of observations, and $p$ is the number of parameters in the model.

The nls function requires a set of starting values to be provided. The starting values were chosen based on biological relevance and were optimized to fit the most genes. The following are the starting values for all parameters used in the models: (i) $b0$, average of expression at 0.01, 0.05, and 0.1 mM cAMP; (ii) $E_{max}$, average of gene expression at 1 mM to 2 mM cAMP; (iii) $n = 2$; (iv) $k = 0.6$; and (v) $m = 1$.

The estimated parameter values and their standard errors extracted from the fit were used to calculate the corresponding RSE for the estimated parameters. The predict.nls function was used to predict expression levels from the fitted function at each cAMP concentration. $R^2_{NLfit}$ values were calculated using custom codes as defined in the previous section.

A gene was considered to have sigmoidal behavior only if its $\sigma^2$ for the Hill's model (equation 1) fit was less than that of the other models, $R^2_{NLfit} > 0.80$ ($P < 0.05$), the slope was $1 \pm 0.2$ and RSE for $E_{max} < 20\%$, and $k < 20\%$. Because no genes are differentially expressed between 0.01 mM cAMP and $\Delta cyaA$, all genes have a $b0$ in the range 0.8- to 1.2-fold change. Because the number of data points around the transition states is few, estimations of $n$ showed higher standard errors than $k$ and $E_{max}$. Hence, for this study, RSE of the estimated $b0$ and $n$ are not used as filters to determine sigmoid genes; 225/305 genes satisfied these criteria.

A gene was considered nonsaturating if the $\sigma^2$ for the fit for the nonsaturating model (equation 2) was lower than any of the other models and its $R^2_{NLfit}$ was greater than 0.7 with a $P$ value of $<0.05$ and slope$_{NLfit}$ of $1 \pm 0.2$. Similarly, for the linear model (equation 3). 28/305 genes satisfied both the models. Very small difference was found in the values of RSE and $R^2_{NLfit}$ for genes that fit both nonsaturating and linear models. Thus, for the ease of interpretability this group is referred to as linear model.

Finally, genes were considered to be nonresponsive if the $\sigma^2$ for the fit to the nonresponsive model (equation 4) was less than that of any other models, $R^2_{NLfit}$ was greater than 0.7, and the $P$ value was less than 0.05. Genes that did not fit any of the categories were analyzed manually. Of the 305 cAMP-activated genes, 225 genes were found to follow Hill's model (HM), 28 genes were best explained by a linear model (LM), and 18 genes failed to respond to extracellular cAMP (NR). We found 34 genes to follow a nonmonotonic inverted U trend in response to increasing cAMP concentrations (NM).

**Cluster analysis.** To cluster the genes based on their trends, dose-response curves of each cAMP-regulated gene were Z score normalized and then partitioned using conventional clustering methods such as $k$-means and hierarchical clustering. Hierarchical clustering was performed using the hclust function from the R Stats package using Pearson distance as the dissimilarity measure and average linkage as the clustering method. The pheatmap package from R was used for the visualization of clustered heatmaps. We wanted to validate the use of Pearson distance as a useful metric for clustering gene

trends using hierarchical clustering. Genes belonging to the same operon (intraoperonic genes) should show high correlation values between the trends they follow compared to nonoperonic genes (interoperonic). We used this comparison as an internal control. Intraoperonic genes showed a higher and tighter spread of Pearson correlation values than interoperonic genes (Fig. S4D), validating that Pearson distance can be used as a metric to cluster our data set.

For $k$-means clustering, the optimum number of clusters was determined using WSS, Gap-stat, and Silhouette methods. The kmeans function from the base R Stats package was used to divide the genes into the desired number of clusters. A PCA plot was used to visualize the clusters formed. Prcomp from the R Stats package was used to calculate the principal components and relative position of each gene.

The clusters formed by $k$-means were used to test whether the presence of outlier genes significantly affected the robustness of the conclusions drawn regarding the Hill's parameters. We selected the genes belonging to the k4-S cluster as a core subset of the Hill's model genes. The k4-S cluster is composed of 93% HM genes (131/142), and 59% of HM genes fall within k4-S (133/225). The remaining 41% of HM genes that also have some superposed characteristics of other models were excluded from k4-S. We found no significant difference between the means of the distributions for $k$, $n$, and $E_{max}$ (Wilcoxon rank-sum test, $P > 0.01$) between k4-S and the total set of HM genes. Similarly, the correlations between $E_{max}$ and CRP (Spearman rank correlation, $\rho = 0.32$, $P = 1.8 \times 10^{-3}$) is maintained (data not shown).

**Data availability.** RNA-seq data are available at Gene Expression Omnibus (GEO) (92) with the accession number GSE202549. ChIP-seq data are available at GEO under accession number GSE92255. All code is available at https://github.com/shwetac09/cAMP_project_codes_2022.git.

## SUPPLEMENTAL MATERIAL

Supplemental material is available online only.

**FIG S1**, PDF file, 0.1 MB.
**FIG S2**, PDF file, 0.4 MB.
**FIG S3**, PDF file, 0.1 MB.
**FIG S4**, PDF file, 0.5 MB.
**FIG S5**, PDF file, 0.4 MB.
**FIG S6**, PDF file, 0.1 MB.
**FIG S7**, PDF file, 0.3 MB.
**FIG S8**, PDF file, 0.1 MB.
**FIG S9**, PDF file, 0.3 MB.
**TABLE S1**, XLSX file, 0.1 MB.

## ACKNOWLEDGMENTS

We thank Sunil Laxman and Sandeep Krishna for reading and offering helpful feedback while writing the manuscript. We also thank Deepa Agashe and Anjana Badrinarayanan for their guidance. We are very grateful to Sachit Daniel and Terence V. Christie for their input and discussions throughout the course of this work. We thank Mohak Sharda and Nitish Malhotra for their comments while proofreading the manuscript. We acknowledge and thank Awadesh Pandit and Tejali Naik at the Next Generation Genomics Facility, NCBS, for providing sequencing services.

This work is supported by a DBT/Wellcome Trust India Alliance Intermediate Fellowship (IA/I/16/2/502711 to A.S.N.S.) and core funding from the Department of Atomic Energy, Government of India (project number 12-R&d-TFR-5.04-0800).

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
