## [Reviewer comments · mSystems]

Understanding the genome-wide transcription response to varying cAMP levels in bacteria using phenomenological models

Shweta Chakraborty, Parul Singh, and Aswin Seshayee

Corresponding Author(s): Aswin Seshayee, Tata Institute of Fundamental Research

Review Timeline:

Submission Date:

September 20, 2022

Accepted:

October 17, 2022

Editor: Richard Notebaart

Reviewer(s): The reviewers have opted to remain anonymous.

Transaction Report:

DOI: <https://doi.org/10.1128/msystems.00900-22>

October 17, 2022

Dr. Aswin Sainarain Seshayee
Tata Institute of Fundamental Research
Bangalore
India

Re: mSystems00900-22 (Understanding the genome-wide transcription response to varying cAMP levels in bacteria using phenomenological models)

Dear Dr. Aswin Sainarain Seshayee:

Your manuscript has been accepted, and I am forwarding it to the ASM Journals Department for publication. For your reference, ASM Journals' address is given below. Before it can be scheduled for publication, your manuscript will be checked by the mSystems production staff to make sure that all elements meet the technical requirements for publication. They will contact you if anything needs to be revised before copyediting and production can begin. Otherwise, you will be notified when your proofs are ready to be viewed.

Publication Fees:

If you would like to submit a potential Featured Image, please email a file and a short legend to mSystems@asmusa.org. Please note that we can only consider images that (i) the authors created or own and (ii) have not been previously published. By submitting, you agree that the image can be used under the same terms as the published article. File requirements: square dimensions (4" x 4"), 300 dpi resolution, RGB colorspace, TIF file format.

We recognize that the video files can become quite large, and so to avoid quality loss ASM suggests sending the video file via <https://www.wetransfer.com/>. When you have a final version of the video and the still ready to share, please send it to mSystems staff at mSystems@asmusa.org.

Sincerely,

Richard Notebaart
Editor, mSystems

Journals Department
Fig. S7: Accept

Fig. S9: Accept

Fig. S2: Accept

Fig. S8: Accept

Supplementary table 1: Accept

Fig. S3: Accept

Fig. S4: Accept

Fig. S1: Accept

Fig. S6: Accept

Fig. S5: Accept